# Outcome benefits of upfront cytoreductive nephrectomy for patients with metastatic renal cell carcinoma: An analysis of the TriNetX database

Gu-Shun Lai[1,2], Jian-Ri Li[1,2,3], Shian-Shiang Wang[1,2,4], Chuan-Shu Chen[1,2], Chun-Kuang Yang[1,2], Chia-Yen Lin[1,2], Sheng-Chun Hung[1,2], Kun-Yuan Chiu[1,2,4], Shun-Fa Yang[1,5]*

1 Institute of Medicine, Chung Shan Medical University, Taichung, Taiwan, 2 Department of Surgery, Division of Urology, Taichung Veterans General Hospital, Taichung, Tawan, 3 Department of Medicine and Nursing, Hungkuang University, Taichung, Taiwan, 4 Department of Applied Chemistry, National Chi Nan University, Nantou, Taiwan, 5 Department of Medical Research, Chung Shan Medical University Hospital, Taichung, Taiwan

* ysf@csmu.edu.tw

**Data Availability Statement:** All relevant data are within the paper and its Supporting Information files.

## Abstract

### Background

The role of upfront cytoreductive nephrectomy remains debatable in the present era of tyrosine kinase inhibitors and immune checkpoint inhibitors. Here, we aimed to evaluate the outcomes of metastatic renal cell carcinoma patients treated with upfront CN and modern systemic therapies.

### Methods

Using the TriNetX network database, we identified patients, in the period from 2008 to 2022, who were diagnosed with metastatic renal cell carcinoma, receiving first-line systemic therapies with tyrosine kinase inhibitors or immune checkpoint inhibitors. Their overall survivals were evaluated using the Kaplan-Meier method as well as multivariable regressions.

### Results

We identified 11,094 patients with metastatic renal cell carcinoma. Of them, 2,914 (43%) patients in the tyrosine kinase inhibitor cohort (n = 6,779), and 1,884 (43.7%) in the immune checkpoint inhibitors cohort (n = 4315) underwent upfront cytoreductive nephrectomy. Those receiving upfront cytoreductive nephrectomy showed survival advantages with either tyrosine kinase inhibitor (Hazard ratio 0.722, 95% Confidence interval 0.67–0.73, p<0.001) or immune checkpoint inhibitors (Hazard ratio 65.1, 95% Confidence interval 0.59–0.71, p<0.001). In multivariable analysis, upfront cytoreductive nephrectomy was a factor for improved OS in both cohorts: tyrosine kinase inhibitors (Hazard ratio 0.623, 95% Confidence interval 0.56–0.694, p<0.001) and immune checkpoint inhibitors cohort (Hazard ratio 0.688, 95% Confidence interval 0.607–0.779, p<0.001).

**Funding:** The author(s) received no specific funding for this work.

**Competing interests:** The authors have declared that no competing interests exist.

## Conclusions

Upfront cytoreductive nephrectomy was associated with an improved overall survival for patients with metastatic renal cell carcinoma receiving either first-line tyrosine kinase inhibitors or immune checkpoint inhibitors. Our results support a clinical role of upfront cytoreductive nephrectomy in the modern era.

## Introduction

Over the past two decades, cytoreductive nephrectomy (CN) has been a standard treatment for patients with metastatic renal cell carcinoma (mRCC). The supporting evidence includes results from several randomized trials showing survival benefits from CN plus interferon treatment compared with interferon therapy alone [1–4]. With the introduction of tyrosine kinase inhibitors (TKI) and immune checkpoint inhibitors (IO), the treatments for mRCC evolved a lot in the past decade [5–8]. Since the role of CN was established before the era of TKI and IO, it is necessary to reassess its influence on oncological outcomes.

In 2018, the randomized phase III trial CARMENA (Cancer du Rein MétastatiqueNéphrectomie et Antiangiogéniques) showed that treatment outcomes in overall survival (OS) with sunitinib alone is not inferior to CN followed by sunitinib for patients with intermediate- or high-risk mRCC [9]. However, subgroup analyses revealed that patients with one risk factor from the International Metastatic Renal Cell Carcinoma Database Consortium (IMDC) have survival benefits with CN. Several studies showed CN plus TKI is associated with improved OS compared with TKI alone for patients with mRCC [10–13]. Recent ASCO guidelines on mRCC also recommended CN as a treatment option for selected patients [14].

Despite the results of the CARMENA trial, CN remains debated in the era of TKI and IO, while it has been used in clinical practice for mRCC patients. Herein, we conducted a retrospective cohort study to assess the role of upfront CN on survival for patients with mRCC receiving either TKI or IO.

## Materials and methods

### Study design, population and outcomes

A retrospective study was performed using TriNetX network, a global database that provides real-world data of ≥250 million people. In this study, we used the US Collaborative Network including 57 healthcare organizations across the US.

We enrolled patients with mRCC and aged ≥18 years old. They received first-line systemic treatment between January 1, 2008 and December 30, 2022. Patients were identified using the International Classification of Diseases, 10th edition, Clinical Modification (ICD-10-CM) codes: ICD-10-CM C64 for malignant neoplasm of kidney, and ICD-10-CM: C78 (lung metastases), C78.7 (liver metastases), C79.3 (brain metastases), or C79.5 (bone metastases) to confirm the diagnosis of distant metastases. The starting date of first-line therapy was set as the index date. Upfront cytoreductive nephrectomy needed to be performed within 3 months before the index date (initiation of first-line therapy). We recognized patients receiving nephrectomy using ICD-10-CM: Z90.5. Comorbidities were identified using ICD-10-CM: I10-I16 for hypertension, ICD-10-CM: I20-I25 for ischemic heart diseases, ICD-10-CM: E08-E13 for diabetes mellitus, and ICD-10-CM: I60-I69 for cerebrovascular disease. The first-line systemic therapies were either TKI (sunitinib, pazopanib, axinitib, or cabozantinib) or IO

(nivolumab, ipilumab, atezolizumab, or pembrolizumab). Patients receiving IO-base combination therapy (IO+TKI) were included into the IO cohort.

The primary outcome was OS. OS was defined as the duration from the index date to the date of death from any cause, or censored at the end of study, whichever happened first.

## Statistical analyses

Patient baseline characteristics, in case of continuous variables, were expressed as mean and standard deviation (SD), and categorical variables, as number and percentage. For evaluating inter-group differences, Student's t test was used for continuous variables, and chi-square test for categorical variables. Survival was evaluated using Kaplan-Meier methodology with a median OS and 95% confidence interval (95% CI), as well as a log-rank test to evaluate inter-group differences in OS. Logistic regression was used to assess various risk factors for death. All analyses were performed on the TriNetX platform. Statistical significance was set at p <0.05.

## Ethics in research

Our study was approved by the institutional review board (IRB) of Taichung Veterans General Hospital (number: SE:22220A). Given information for patient identification was not provided on the TriNetX platform, the IRB waived the requirement for informed consent.

## Results

### Baseline characteristics

Patient characteristics are shown in Table 1. In total, we identified 11,094 patients with mRCC with most of the patients being white. Of them, 6,779 received TKI therapy (the TKI cohort), and 2,914(43%) of them had CN. Also, 4,315 of these patients received IO therapy (the IO cohort), and 1,884 (43.7%) had CN. In the TKI cohort, the majority of patients receive sunitinib (n = 1,984, 29.2%) and pazopanib (n = 1,947, 28.7%). In the IO cohort, the most frequently used treatments were pembrolizumab (n = 1,558, 36.1%) and nivolizumab (n = 1140, 26.4%). Pei chart of races and first-line systemic therapies in the TKI and IO cohorts was illustrated in Fig 1.

In the TKI cohort, patients receiving CN had significantly more distant metastases and comorbidities when compared with those without CN (all with p<0.001). Similarly, in the IO cohort, patients receiving CN had more instances of hypertension (p<0.001), diabetes mellitus (p = 0.0012), ischemic heart disease (p<0.001), and lung metastasis (p<0.001), while bone metastasis occurred more frequently for patients not receiving CN (p<0.001) (Fig 2). The Eastern cooperative oncology group (ECOG) performance status was better for patients receiving CN in both TKI and IO cohorts (TKI cohort: p = 0.016, IO cohort: p = 0.044).

### Outcomes

In the TKI cohort, the median follow-up time was 30.1 months, and in the IO cohort, this was 28.8 months. By the end of this study (December, 2022), 3,540 (52.2%) patients in the TKI cohort reached primary end point (deaths), and 2,086 (48.3%) in the IO cohort.

In the TKI cohort, for patients with CN, their survival probability at the 12th month was 73.9% [95% confidence interval (CI) 72.1–75.5] compared with 64.9% (95% CI 63.2–66.5) for those without CN. In the IO cohort, for patients with CN, their survival probability at the 12th month was 71.4% (95% CI 69.1–73.5), compared with 60.3% (95% CI 57.8–62.1) for those without CN. In the TKI cohort, their median OS was 38.3 months for patients with CN, and

**Table 1. Baseline characteristics for patients with metastatic renal cell carcinoma receiving first-line tyrosine kinase inhibitors or immune checkpoint inhibitors.**

| | TKI with CN n = 2914 | TKI without CN n = 3865 | P | IO with CN n = 1884 | IO without CN n = 2431 | P |
|---|---|---|---|---|---|---|
| Age, years, mean (SD) | 63.3 (11.2) | 62.9 (12) | 0.0607 | 66.2 (11.7) | 65.9 (12.7) | 0.534 |
| Sex, male (%) | 2063 (71) | 2666 (69) | 0.106 | 1322 (70) | 1641 (68) | 0.061 |
| Race, n (%) | | | | | | |
| White | 2363(81) | 2781 (72) | <0.001 | 1478 (79) | 1880 (77) | 0.359 |
| Black | 202 (7) | 274 (10) | <0.001 | 102 (5) | 204 (8) | 0.0002 |
| Asian | 58 (2) | 86 (2) | 0.507 | 66(4) | 78 (3) | 0.0593 |
| Others/unknown | 291(9) | 724 (19) | <0.001 | 93 (16) | 269 (11) | 0.2163 |
| BMI, mean (SD) | 29.2 (6.48) | 28.6 (6.37) | 0.009 | 28.6 (6.25) | 28 (6.24) | 0.0152 |
| ECOG, mean (SD) | 0.562 (0.666) | 0.943 (0.802) | 0.0106 | 0.596(0.64) | 0.781(0.72) | 0.044 |
| Metastatic site, n (%) | | | | | | |
| Lung | 1370 (47) | 859 (22) | <0.001 | 801 (43) | 774 (32) | <0.001 |
| Liver | 457 (16) | 391 (10) | <0.001 | 267 (14) | 361 (15) | 0.531 |
| Bone | 932 (32) | 883 (23) | <0.001 | 496 (26) | 764 (31) | 0.0003 |
| Brain | 368 (13) | 291 (8) | <0.001 | 191 (10) | 272 (11) | 0.2687 |
| Comorbidity, n (%) | | | | | | |
| Diabetes Mellitus | 919 (32) | 766 (20) | <0.001 | 629 (33) | 700 (29) | 0.0012 |
| Hypertension | 2053 (70) | 1707(44) | <0.001 | 1389 (74) | 1492 (61) | <0.001 |
| Cerebrovascular disease | 416 (14) | 317 (8) | <0.001 | 333 (18) | 391 (16) | 0.1653 |
| Ischemic heart disease | 760(26) | 529 (14) | <0.001 | 727 (39) | 630 (26) | <0.001 |
| Types of TKI, n (%) | | | | | | |
| Sunitinib | 866 (30) | 1118 (29) | | | | |
| Pazopanib | 867 (30) | 1080 (28) | | | | |
| Cabozatinib | 558 (19) | 728 (19) | | | | |
| Axitinib | 372 (13) | 584 (15) | | | | |
| Unknowns | 251 (9) | 355 (9) | | | | |
| Types of IO, n (%) | | | | | | |
| Pembrolizumab | | | | 636 (34) | 922 (38) | |
| Ipilimumab + Nivolumab | | | | 550 (29) | 536 (22) | |
| Nivolumab | | | | 521 (28) | 619 (25) | |
| Atezolizumab | | | | 119 (6) | 172 (7) | |
| Unknowns | | | | 58 (3) | 182 (7) | |

BMI: Body mass index; CN: Cytoreductive nephrectomy; ECOG, eastern cooperative oncology Group; IO: Immune checkpoint inhibitor; SD, standard deviation; TKI: Tyrosine kinase inhibitor.

23.3 months for those without CN. In the IO cohort, their median OS was 40.5 months for patients with CN, and 19.1 months for those without CN. Patients undergoing CN had survival benefits in OS for both TKI [Hazard ratio (HR) 0.722, 95% CI 0.67–0.73, p<0.001] and IO (HR 65.1, 95% CI 0.59–0.71, p<0.001) cohorts (Fig 3). For patients receiving CN, there was no significant difference in OS between TKI and IO cohorts.

Based on multivariable logistic regression analyses, CN was associated with a reduced risk of death in both TKI (HR 0.623, 95% CI 0.56–0.694, p<0.001) and IO (HR 0.688, 95% CI 0.607–0.779, p<0.001) cohorts (Table 2 and Fig 4).

We performed subgroup analyses and found similar results. Patients with mRCC experienced survival benefits from CN treated with either TKI monotherapy (HR 0.643, 95% CI 0.598–0.692, p<0.001), IO monotherapy (HR 0.675, 95% CI 0.607–0.75, p<0.001),

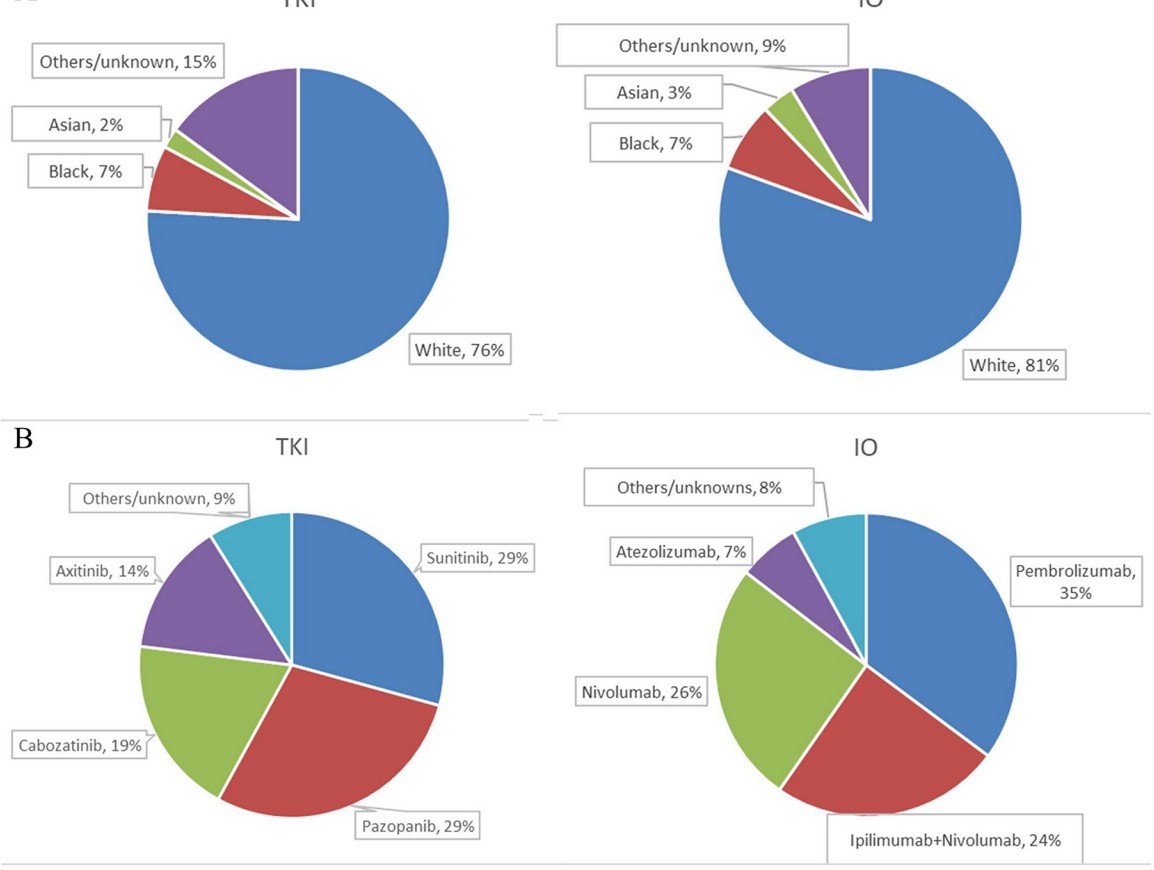

**Fig 1.** Pie charts of (A) races and (B) first-line systemic therapies for patient with metastatic renal cell carcinoma in the TKI and IO cohorts. IO: Immuno-oncology; TKI: Tyrosin kinase inhibitor.

Ipilimumab + Nivolumab (HR 0.491, 95% CI 0.422–0.571, p<0.001), or Axitinib + Pembrolizumab (HR 0.461, 95% CI 0.369–0.575, p<0.001) (Fig 5).

## Discussion

In the present study, we conducted a retrospective cohort study on the TriNetX platform to evaluate the benefits of upfront CN for patients with mRCC receiving first-line systemic therapy with either TKI or IO. We found that patients receiving upfront CN, compared with those without CN, were associated with improved OS.

CN was historically the standard care option for patients with mRCC. Its evidence is based on several trials reporting survival benefits of surgical intervention in the era of cytokines [1–4]. With the advent of targeted therapies and immune checkpoint inhibitors, several studies reported survival advantages of these new systemic treatments over traditional cytokines [5–8]. Given the rapid evolution of these novel and more efficient agents, the role of CN has become controversial. CARMENA, a phase 3, randomized trial on patients with mRCC, reported that sunitinib alone is not inferior in OS when compared with CN followed by sunitinib [9]. SURTIME is a randomized trial, which demonstrated a survival advantage of deferred CN compared with immediate CN, indicating that surgical intervention could be an option for patients with objective response to sunitinib [10]. Nevertheless, a subgroup analysis in the

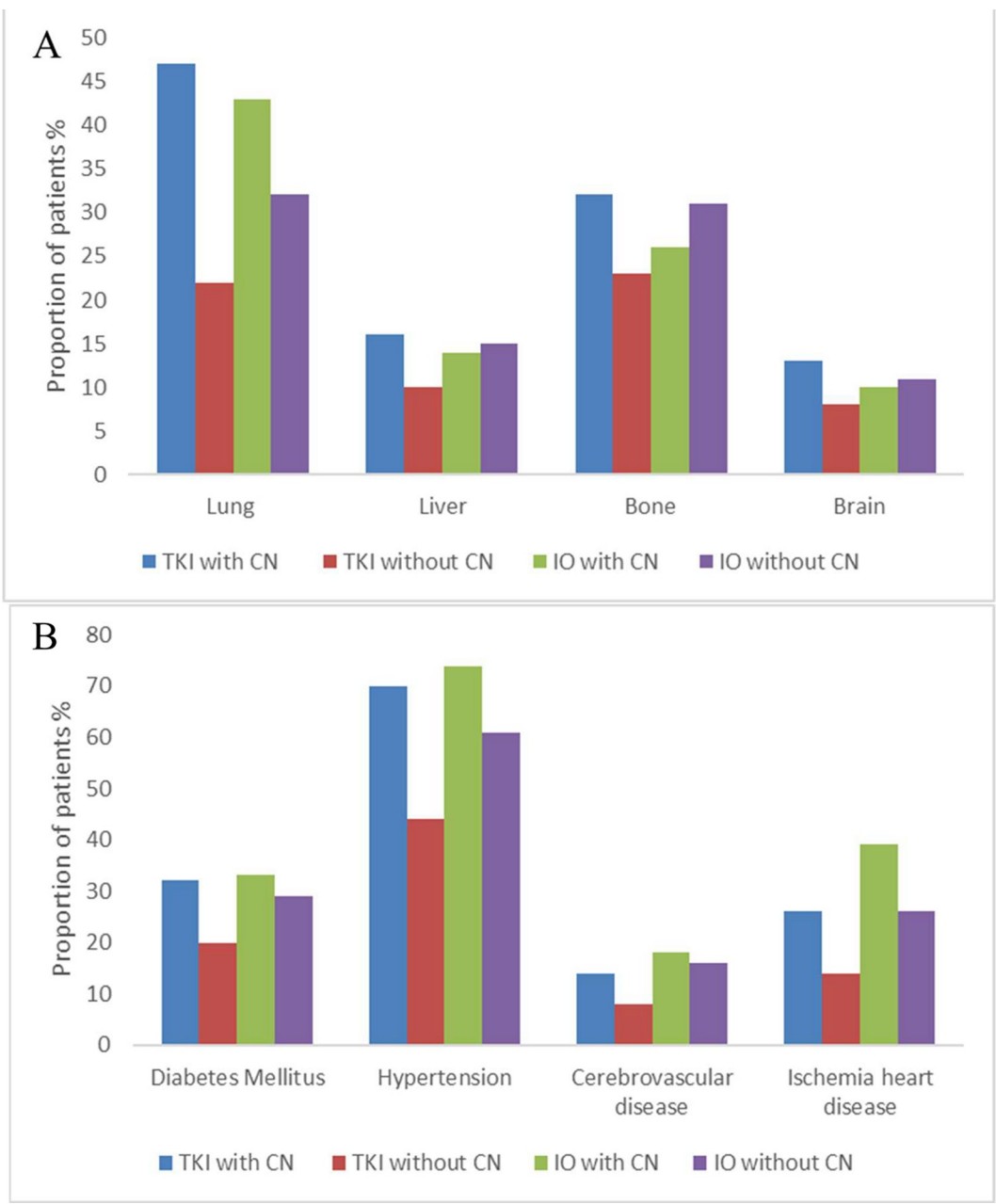

**Fig 2.** Bar graph of (A) sites of distant metastases and (B) comorbidities for patients with metastatic renal cell carcinoma in the TKI with/without CN and IO with/without CN cohorts. CN: Cytoreductive nephrectomy; IO: Immune checkpoint inhibitors; TKI: Tyrosin kinase inhibitor.

CARMENA trial reported that patients with one IMDC risk factor have survival benefit from CN, and multiple studies also supported the role of CN in the modern era [11–17]. In this study, we found that patients undergoing upfront CN were associated with better OS in both TKI and IO cohorts [18,19]. One hypothesis of the underlying mechanisms is that primary RCC releases cytokines to stimulate inflammation, and they also reducing immune responses against the tumor [20,21]. CN reduces cytokines and prevents metastatic tumors from progression. The potential immune modulation effects of surgery may be further aggravated in patients treated with IO therapies.

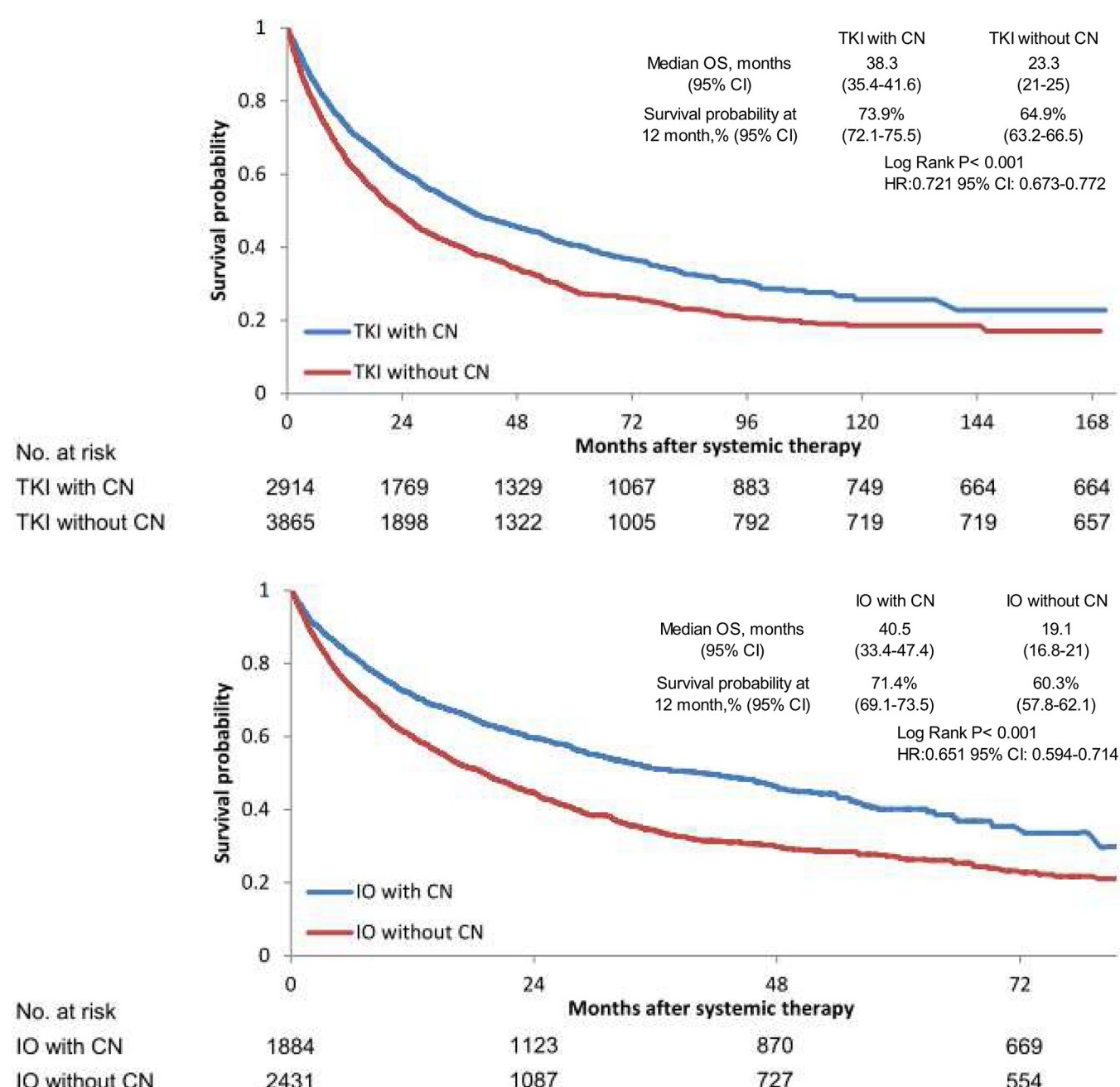

**Fig 3.** Kaplan-Meier survival curves of overall survival for patients with metastatic renal cell carcinoma treated with (A) TKI with/without CN and (B) IO with/without CN. CI: Confidence Interval; CN: Cytoreductive nephrectomy; HR: Hazard ration; IO: Immune checkpoint inhibitor; TKI: Tyrosin kinase inhibitor.

According to previous studies, patients with fewer sites of metastases and better performance status have survival benefit from surgical intervention [9,13,22,23]. In this study, we found that patients receiving CN have better ECOG performance. Interestingly, they had more incidences of distant metastases and co-morbidities. Multiple variables analyses confirmed that fewer metastases and CN were associated with better OS. Results indicated that patients with more distant metastases and co-morbidities may still get survival benefit from surgery.

**Table 2. Multivariable analysis for overall survival in patients with metastatic renal cell carcinoma receiving first-line tyrosine kinase inhibitors or immune checkpoint inhibitors.**

| | TKI | | | | IO | | | |
|---|---|---|---|---|---|---|---|---|
| | HR | 95% CI | | P | HR | 95% CI | | P |
| Age at index date | 1.007 | 1.003 | 1.012 | 0.001 | 1.018 | 1.013 | 1.023 | <0.001 |
| Male Gender (Male/Female) | 1.125 | 1.015 | 1.247 | 0.024 | 1.089 | 0.962 | 1.232 | 0.177 |
| Upfront cytoreductive nephrectomy | 0.623 | 0.56 | 0.694 | <0.001 | 0.688 | 0.607 | 0.779 | <0.001 |
| Bone metastases | 1.351 | 1.215 | 1.503 | <0.001 | 1.396 | 1.227 | 1.587 | <0.001 |
| Brain metastases | 1.373 | 1.165 | 1.617 | <0.001 | 1.221 | 1.006 | 1.481 | 0.043 |
| Liver metastases | 1.596 | 1.38 | 1.847 | <0.001 | 1.857 | 1.57 | 2.196 | <0.001 |
| Lung metastases | 1.228 | 1.108 | 1.361 | <0.001 | 1.223 | 1.082 | 1.382 | 0.001 |
| ECOG performance status (≥2/0-1) | 1.171 | 0.56 | 3.625 | 0.785 | 1.021 | 0.52 | 2.005 | 0.951 |
| Clear cell/non-clear cell histology | 1.132 | 0.972 | 1.317 | 0.111 | 0.788 | 0.627 | 0.988 | 0.039 |
| Hypertension | 1.074 | 0.963 | 1.197 | 0.202 | 1.006 | 0.877 | 1.155 | 0.932 |
| Diabetes mellitus | 1.048 | 0.931 | 1.179 | 0.436 | 1.02 | 0.893 | 1.165 | 0.772 |
| Cerebrovascular disease | 1.365 | 1.161 | 1.605 | <0.001 | 1.381 | 1.173 | 1.632 | <0.001 |
| Ischemic heart disease | 1.128 | 0.99 | 1.285 | 0.071 | 1.202 | 1.046 | 1.379 | 0.009 |

ECOG: Eastern cooperative oncology group; HR: Hazard ratio; IO: Immune checkpoint inhibitor; TKI: Tyrosin kinase inhibitor; CI: Confidence interval.

The discrepancy of these findings highlighted the survival advantages of CN in mRCC, suggesting that multiple factors should be considered for clinicians in evaluating these patients.

There are some limitations of our study. First, its retrospective design and non-randomization are subject to selection bias. Second, some patient information and statistics analyses were not available on the platform, such as IMDC risk stratification and Cox regression analysis. Despite these limitations, our study involved a large population in a real-world setting. Our findings provide clinicians some useful information in the management of these patients while awaiting the results from the ongoing trials that assess the role of CN in the era of TKI and IO.

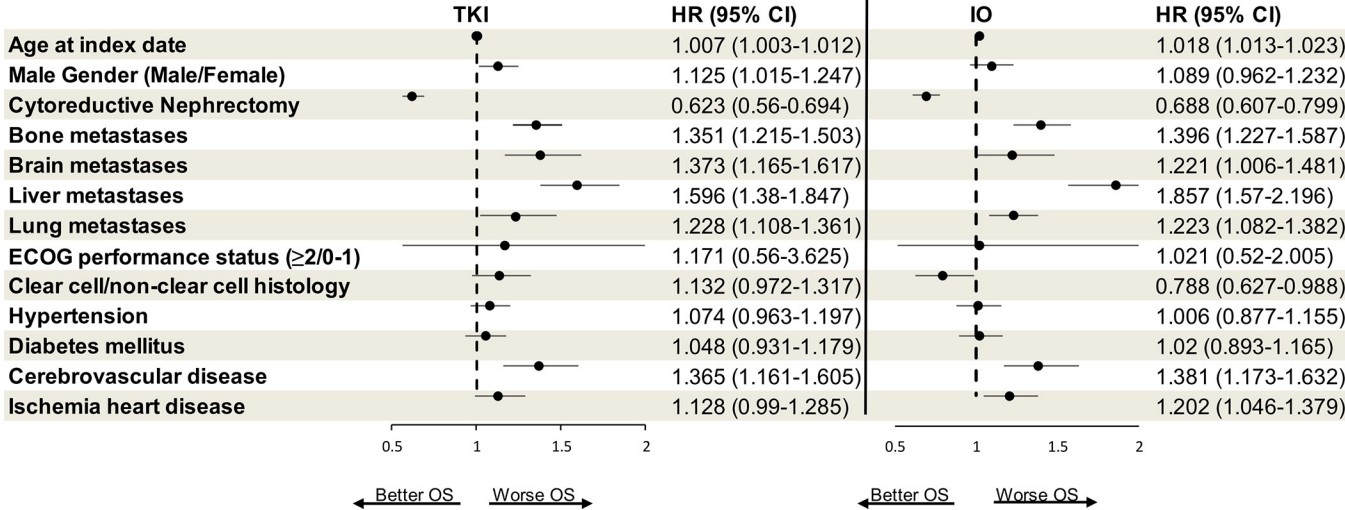

**Fig 4. Forest plot of multivariable analysis for overall survival in patients with metastatic renal cell carcinoma treated with TKI and IO.** CI: Confidence interval; HR: Hazard ration; IO: Immune checkpoint inhibitors; OS: Overall survival; TKI: Tyrosin kinase inhibitor.

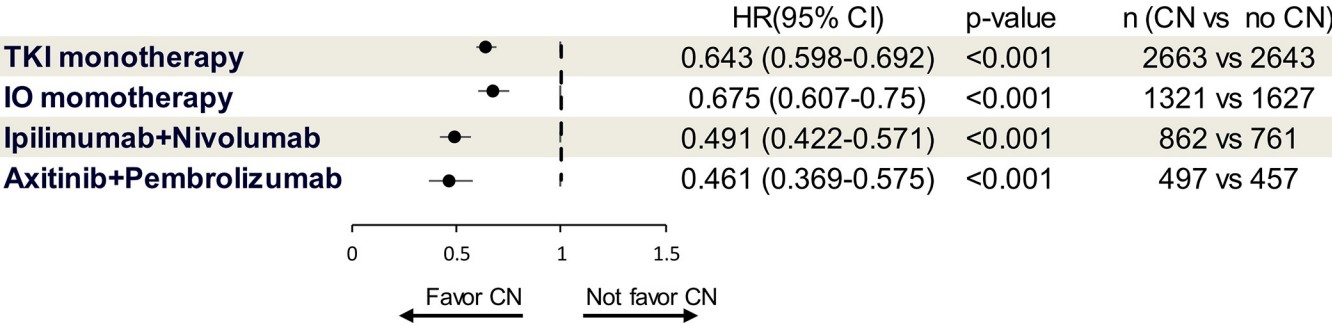

**Fig 5. Subgroup analysis of overall survival for patients with metastatic renal cell carcinoma treated with TKI momotherapy, IO monotherapy, Ipilimumab+Nivolumab, and Axitinib + Pembrolizumab.** CI: Confidence interval; CN: Cytoreductive nephrectomy; HR: Hazard ratio; IO: Immune checkpoint inhibitors; OS: Overall survival; TKI: Tyrosin kinase inhibitor.

## Conclusion

Results of our study supported the OS benefits of upfront CN for mRCC patients in the modern TKI and IO era. Before reports emerge from prospective and randomized trials, our findings are helpful for clinicians in treating these patients.

## Supporting information

**S1 File. Values for Kaplan-Meier survival analysis in patients with TKI treatment.** (XLSX)

**S2 File. Values for Kaplan-Meier survival analysis in patients with IO treatment.** (XLSX)

## Acknowledgments

This study was based on the data from the TriNetX network.

## Author Contributions

**Conceptualization:** Gu-Shun Lai, Jian-Ri Li, Shun-Fa Yang.

**Data curation:** Gu-Shun Lai.

**Formal analysis:** Gu-Shun Lai.

**Supervision:** Jian-Ri Li, Shun-Fa Yang.

**Writing – original draft:** Gu-Shun Lai.

**Writing – review & editing:** Gu-Shun Lai, Jian-Ri Li, Shian-Shiang Wang, Chuan-Shu Chen, Chun-Kuang Yang, Chia-Yen Lin, Sheng-Chun Hung, Kun-Yuan Chiu, Shun-Fa Yang.

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
