## [Decision Letter · Decision Letter 0]

11 Jan 2024

PONE-D-23-31046Outcome benefits of upfront cytoreductive nephrectomy for patients with metastatic renal cell carcinoma: an analysis of the TriNetX databasePLOS ONE

Dear Dr. Lai,

Thank you for submitting your manuscript to PLOS ONE. After careful consideration, we feel that it has merit but does not fully meet PLOS ONE’s publication criteria as it currently stands. Therefore, we invite you to submit a revised version of the manuscript that addresses the points raised during the review process.

We look forward to receiving your revised manuscript.

Kind regards,

Mirosława Püsküllüoğlu, MD, PhD

Academic Editor

PLOS ONE

5. Please remove your figures from within your manuscript file, leaving only the individual TIFF/EPS image files, uploaded separately. These will be automatically included in the reviewers’ PDF.

Reviewers' comments:

Reviewer's Responses to Questions

**Comments to the Author**

1. Is the manuscript technically sound, and do the data support the conclusions?

Reviewer #1: Partly

Reviewer #2: No

Reviewer #3: Yes

2. Has the statistical analysis been performed appropriately and rigorously? 

Reviewer #1: Yes

Reviewer #2: No

Reviewer #3: Yes

3. Have the authors made all data underlying the findings in their manuscript fully available?

Reviewer #1: Yes

Reviewer #2: No

Reviewer #3: Yes

4. Is the manuscript presented in an intelligible fashion and written in standard English?

Reviewer #1: No

Reviewer #2: Yes

Reviewer #3: Yes

5. Review Comments to the Author

Reviewer #1: A major shortcoming of the manuscript is graphical representation of the scientific data. I strongly suggest inclusion of graphs to match both the standard of the journal and also global reporting criteria.

Reviewer #2: I read the manuscript with interest and it addresses an important issue about the role of CN in modern era. There are however serious issues with technical aspects, especially the analysis.

the authors need to clearly mention in the methods section which all variables were used for multivariable analysis. Also, IMDC or other composite measures of disease burden (if present) are welcome in the analysis. There is mention of logistic regression used for analysis of death. Death is a time to event data and cox proportional hazard regression is more appropriate design. if time to event data is available to draw OS curves, then cox PH analysis should be feasible. please also give data on number at risk below the OS curves. Try to see if Charlson comorbidity index score can be assigned to make a composite score for comorbidities.

Reviewer #3: Well written study examining CN for RCC using the TriNetX database.

Major Points:

- Consider citing the recent ASCO guideline on metastatic RCC as it specifically addresses upfront CN.

Management of Metastatic Clear Cell Renal Cell Carcinoma: ASCO Guideline.

Rathmell WK, Rumble RB, Van Veldhuizen PJ, Al-Ahmadie H, Emamekhoo H, Hauke RJ, Louie AV, Milowsky MI, Molina AM, Rose TL, Siva S, Zaorsky NG, Zhang T, Qamar R, Kungel TM, Lewis B, Singer EA.

J Clin Oncol. 2022 Sep 1;40(25):2957-2995. doi: 10.1200/JCO.22.00868. Epub 2022 Jun 21.

PMID: 35728020

- Consider performing separate analyses for TKI monotherapy, IO monotherapy, and doublet therapy (IO/IO and IO/TKI) instead of combining IO and doublets.

Minor Points:

- None

6. PLOS authors have the option to publish the peer review history of their article (what does this mean?). If published, this will include your full peer review and any attached files.

Reviewer #1: No

Reviewer #2: **Yes: **Jiten Jaipuria

Reviewer #3: No

---

## [Author Response · Author response to Decision Letter 0]

20 Jan 2024

Reviewer #1: A major shortcoming of the manuscript is graphical representation of the scientific data. I strongly suggest inclusion of graphs to match both the standard of the journal and also global reporting criteria.

We included number of patients and related data to present the outcomes in the survival curves (Fig 3). Please check it in the revised manuscript.

Reviewer #2: I read the manuscript with interest and it addresses an important issue about the role of CN in modern era. There are however serious issues with technical aspects, especially the analysis.

the authors need to clearly mention in the methods section which all variables were used for multivariable analysis. Also, IMDC or other composite measures of disease burden (if present) are welcome in the analysis. There is mention of logistic regression used for analysis of death. Death is a time to event data and cox proportional hazard regression is more appropriate design. if time to event data is available to draw OS curves, then cox PH analysis should be feasible. please also give data on number at risk below the OS curves. Try to see if Charlson comorbidity index score can be assigned to make a composite score for comorbidities.

All variables and related ICD codes used in the multivariable analysis were added in the method section. Number at risk were added below the survival curves (Fig 3). Please check it in the revised manuscript. We regretted that time to event, Charlson comorbidity index and IMDC data are currently not available on the TriNetX database. 

Reviewer #3: Well written study examining CN for RCC using the TriNetX database.

Thanks for the comments. 

Major Points:

- Consider citing the recent ASCO guideline on metastatic RCC as it specifically addresses upfront CN.

Management of Metastatic Clear Cell Renal Cell Carcinoma: ASCO Guideline.

Rathmell WK, Rumble RB, Van Veldhuizen PJ, Al-Ahmadie H, Emamekhoo H, Hauke RJ, Louie AV, Milowsky MI, Molina AM, Rose TL, Siva S, Zaorsky NG, Zhang T, Qamar R, Kungel TM, Lewis B, Singer EA.

J Clin Oncol. 2022 Sep 1;40(25):2957-2995. doi: 10.1200/JCO.22.00868. Epub 2022 Jun 21.

PMID: 35728020

The journal regarding recent ASCO guideline on metastatic RCC was cited in the revised manuscript (reference No. 14). Please check it.

- Consider performing separate analyses for TKI monotherapy, IO monotherapy, and doublet therapy (IO/IO and IO/TKI) instead of combining IO and doublets.

Subgroup analyses for TKI monotherapy, IO monotherapy, and doublet therapy (IO/IO with Ipilimumab + Nivolumab and IO/TKI with Axitinib + Pembrolizumab) were performed and added in the revised manuscript (Fig 5). Please check it.

Review Comments on “Outcome benefits of upfront cytoreductive nephrectomy for patients with metastatic renal cell carcinoma: an analysis of the TrinetX database” 

Overall comments: 

The authors report potential clinical implications and survival of cytoreductive nephrotomy in patients. The manuscript is a well-written one and describes the limitations of the study too. However, I do have the following suggestions as it would help the reader to understand the fi ndings more conveniently, specifi cally tables 1 and 2, than how it is now. Because it is difficult now to go through the data and the graphs are crucial to be a part of the scientific data presentation in the manuscript, also required to match the standards of the journal. 

1. From Table 1, it is suggested that addition of pie-charts to show the points mentioned in the chart like age, sex, ethnicity, organs, etc. in both TKI and IO, is done. This way it will depict the data in a better presentable manner. 

2. From Table 1, similarly a bar graph representation with the statistics included for the BMI and co-morbidities is what is needed. 

3. A volcano plot is also preferred to be added to show the distribution of the experimental groups amongst the individuals studied, with and without the treatments. 

4. The above-mentioned points need to be addressed for table 2 too.

Pei-charts and bar graphs to report the scientific data in the Table 1 were included in the revised manuscript (Fig 1 and 2). We used a Forest plot to present the data in Table 2 (Fig 4). Please check it.

---

## [Decision Letter · Decision Letter 1]

6 Feb 2024

Outcome benefits of upfront cytoreductive nephrectomy for patients with metastatic renal cell carcinoma: an analysis of the TriNetX database

PONE-D-23-31046R1

Dear Dr. Lai,

We’re pleased to inform you that your manuscript has been judged scientifically suitable for publication and will be formally accepted for publication once it meets all outstanding technical requirements.

Kind regards,

Mirosława Püsküllüoğlu, MD, PhD

Academic Editor

PLOS ONE

Additional Editor Comments (optional):

Reviewers' comments:

Reviewer's Responses to Questions

**Comments to the Author**

1. If the authors have adequately addressed your comments raised in a previous round of review and you feel that this manuscript is now acceptable for publication, you may indicate that here to bypass the “Comments to the Author” section, enter your conflict of interest statement in the “Confidential to Editor” section, and submit your "Accept" recommendation.

Reviewer #1: All comments have been addressed

Reviewer #3: All comments have been addressed

2. Is the manuscript technically sound, and do the data support the conclusions?

Reviewer #1: Yes

Reviewer #3: Yes

3. Has the statistical analysis been performed appropriately and rigorously? 

Reviewer #1: Yes

Reviewer #3: Yes

4. Have the authors made all data underlying the findings in their manuscript fully available?

Reviewer #1: Yes

Reviewer #3: Yes

5. Is the manuscript presented in an intelligible fashion and written in standard English?

Reviewer #1: Yes

Reviewer #3: Yes

6. Review Comments to the Author

Reviewer #1: According to me, most of the concerns that were raised by the reviewers, has been addressed. The data, with the addition of graphs, has a better representation in a scientific way. Although I have mentioned about some minor changes regarding the formatting issues.

Reviewer #3: All queries have been addressed. No additional queries. I recommend publication.

Thank yo for the opportunity to review.

7. PLOS authors have the option to publish the peer review history of their article (what does this mean?). If published, this will include your full peer review and any attached files.

Reviewer #1: No

Reviewer #3: No
